# Molecular Xenomonitoring (MX) allows real-time surveillance of West Nile and Usutu virus in mosquito populations

**Clément Bigeard[1,2], Laura Pezzi[3,4], Raphaelle Klitting[3,4], Nazli Ayhan[3,4], Grégory L'Ambert[5], Nicolas Gomez[6], Géraldine Piorkowski[4], Rayane Amaral[4], Guillaume André Durand[3,4], Agathe M. G. Colmant[4], Cynthia Giraud[4], Katia Ramiara[1], Camille Migné[2], Gilda Grard[3,4], Thierry Touzet[7], Stéphan Zientara[2], Rémi Charrel[4], Gaëlle Gonzalez[2], Alexandre Duvignaud[1], Denis Malvy[1], Xavier de Lamballerie[3,4], Albin Fontaine[4,8] ***

**1** Department of Infectious Diseases and Tropical Medicine, CHU Bordeaux, France; National Institute for Health and Medical Research (INSERM) UMR 1219, Research Institute for Sustainable Development (IRD) EMR 271, Bordeaux Population Health Research Centre, University of Bordeaux, Bordeaux, France, **2** ANSES, INRAE, Ecole Nationale Vétérinaire d'Alfort, UMR Virologie, Laboratoire de Santé Animale, Maisons-Alfort, France, **3** Centre National de Référence des Arbovirus, Inserm-IRBA, Marseille, France, **4** Unité des Virus Émergents (UVE: Aix-Marseille Univ, Università di Corsica, IRD 190, Inserm 1207, IRBA), France, **5** Entente interdépartementale pour la démoustication du littoral méditerranéen (EID Méditerranée), Montpellier, France, **6** Unité de Parasitologie et Entomologie, Département Microbiologie et maladies infectieuses, Institut de Recherche Biomédicale des Armées (IRBA), Marseille, France, **7** Direction Régional de l'Alimentation de l'Agriculture et de la Forêt (DRAAF) de Nouvelle-Aquitaine, **8** Institut de Recherche Biomédicale des Armées (IRBA), Unité de virologie, Marseille, France

* albin.FONTAINE@univ-amu.fr

**Data Availability Statement:** Data are available under the NCBI BioProject number PRJNA1085973. Analysis files are available at

## Abstract

West Nile Virus (WNV) and Usutu virus (USUV) circulate through complex cryptic transmission cycles involving mosquitoes as vectors, birds as amplifying hosts and several mammal species as dead-end hosts. Both viruses can be transmitted to humans through mosquito bites, which can lead to neuroinvasive and potentially fatal disease. Notably, WNV can also be transmitted through blood donations and organ transplants. The high proportion of asymptomatic infections caused by these viruses and their cryptic enzootic circulation make their early detection in the environment challenging. Viral surveillance in France still heavily relies on human and animal surveillance, *i.e.* late indicators of viral circulation. Entomological surveillance is a method of choice for identifying virus circulation ahead of the first human and animal cases and to reveal their genetic identity, but performing molecular screening of vectors is expensive, and time-consuming. Here we show substantial WNV and USUV co-circulation in Atlantic seaboard of France between July and August 2023 using a non-invasive MX (Molecular Xenomonitoring) method that use trapped mosquito excreta. MX offers significant advantages over traditional entomological surveillance: it is cost-effective and efficient, enabling viral RNA screening from a community of trapped mosquitoes via their excreta, which can be transported at room temperature. Additionally, MX extends the longevity of trapped mosquitoes, enhancing virus detection and simplifying logistics, and is easy to implement without requiring specialized skills. At the crossroads between entomological and environmental surveillance, MX can detect the circulation of

https://github.com/rklitting/WNV_USUV_
NouvelleAquitaine_2023.

**Funding:** This study received funding from the Direction Générale de l'Armement (PDH-2-NBC-5-B-2212) AB and ARBOGEN (funded by MSDAVENIR) RK. This work was also supported by the French Ministry of Agriculture (CB) and ANRS MIE (DM). The funders had no role in study design, data collection and analysis, decision to publish, or preparation of the manuscript.

**Competing interests:** The authors have declared that no competing interests exist.

zoonotic pathogens in the environment before cases are observed in humans and horses, enabling the timely alerts to health policy makers, allowing them to take suitable control measures.

## Author summary

We described the emergence of West Nile Virus (WNV) in a new region of France and its co-circulation with Usutu Virus (USUV), another zoonotic mosquito-borne virus. Although infections with both viruses can often be asymptomatic, they have the potential to cause severe and potentially fatal illnesses affecting the central nervous system, especially in vulnerable human populations. Our goal was to address the challenges of early detection and efficient surveillance of these viruses. We used an efficient and cost-effective method (ie. Molecular Xenomonitoring (MX)) that tests excreta shed by a community of trapped mosquito for viral RNA to reveal the presence of a new focus of arbovirus circulation. Traditional surveillance methods, which rely heavily on monitoring human and animal infections, provide late indicators of viral presence. Early detection is crucial for enabling timely public health interventions to protect vulnerable groups, such as implementing systematic viral RNA screening of blood and organ donors. MX simplifies logistics, requires no specialized skills, and enables early detection of virus circulation, facilitating timely public health interventions. Our findings demonstrated MX's effectiveness in identifying significant WNV and USUV circulation in a newly affected area of France, providing crucial genomic data and improving virus monitoring efforts.

## Introduction

Native to sub-Saharan Africa, WNVs can be classified into eight phylogenetic lineages, with lineages 1 and 2 causing human disease [1]. Spread globally through bird migrations and human activities, WNV significantly impacted public and animal health outside Africa. Introduced to the USA in 1999, WNV caused the largest recorded outbreaks of neuroinvasive arboviral disease, with tens of thousands of cases and several thousand deaths [2]. In Europe, WNV appeared in the 1960s [3], with lineage 1a [4] causing sporadic cases until the emergence of lineage 2 in 2004, which led to larger outbreaks [5]. In France, WNV lineages 1 and 2 circulate mainly in the Mediterranean basin, occasionally causing infections in humans, equids, and birds (Box 1).

### Box 1: Emergence timeline of WNV and USUV

**1960** –Emergence of WNV lineage 1 in Europe, including in Southern France [3].

**1999** –Emergence of WNV lineage 1 in the United States of America. The virus has since spread southwards across the continent [2].

**2000**–76 WNV cases in Equidae in Camargue, 21 deaths (Provence-Alpes-Côte d'Azur region) [12].

**1996/2001** –Emergence of USUV in Europe [6].

**2001/2002** –Low level of WNV activity in Camargue as reported in sentinel birds (Provence-Alpes-Côte d'Azur region) [13].

**2003**–7 and 4 WNV cases in Human and Equidae, respectively, in the Var department (Provence-Alpes-Côte d'Azur region) [13].

**2004** –Emergence of WNV lineage 2 in Europe [5]. 37 suspected WNV cases in Equidae in Camargue (Provence-Alpes-Côte d'Azur region) [13].

**2006**–4 WNV cases in Equidae in Pyrénées-Orientales department (Occitanie region) [14].

**2008** –Large WNV outbreaks in three Italian Northern Regions (Emilia Romagna, Veneto, Lombardy) with 794 cases of WNV infections in Equidae, several WNV infection detected in birds (magpies, carrion crows, and rock pigeons) and 9 WNV cases in Human [14].

**2009/2015** –Emergence of USUV in France [8–10].

**2015**–49 WNV cases in Equidae in Camargue and Hérault department (Occitanie region)[15] and 1 WNV human case in Gard department (Occitanie region) [16].

**2017**–2 and 1 WNV cases in Human and Equidae, respectively, in Gard department (Occitanie region) [16].

**2018** –High number of WNV and USUV human and animal cases in Europe. 26 and 13 WNV cases in Human, Equidae, respectively [16]. In the avifauna: 4 WNV cases in northern goshawks, 1 in common buzzard and one in long-eared owl [16] in Corsica and Alpes-Maritimes (Provence-Alpes-Côte d'Azur region). USUV is detected in a Lapland Owl in the Gironde (Nouvelle-Aquitaine region) department and in a blackbird in the Charente department (Nouvelle-Aquitaine region).

**2019**–9 WNV cases in Equidae in Camargue (Provence-Alpes-Côte d'Azur region) [16].

**2022, September**–USUV is detected in a Lapland Owl in the Dordogne department (Nouvelle-Aquitaine region).

**2022, October**–First evidence of WNV circulation on the Atlantic coast of France (3 symptomatic horses) and first human case of USUV.

**2023, July 16th**–First WNV human case in the Atlantic coast of France.

**2023, July 21st**–First USUV human case in Nouvelle-Aquitaine in 2023.

**2023, July 24th**–MX revealed the circulation of both WNV and USUV in Nouvelle-Aquitaine in 2023.

**2023, August 4th**–First WNV equine case in Nouvelle-Aquitaine in 2023.

**2023, August**–USUV is detected in a blackbird in the Charente-Maritime department (Nouvelle-Aquitaine region).

**2023, September**–USUV and WNV are detected (co-infection) in a wood pigeon in the Charente department (Nouvelle-Aquitaine region).

**2023, November**–USUV is detected in a Lapland Owl in the Dordogne department (Nouvelle-Aquitaine region).

Usutu virus (USUV) was first detected in Europe in 2001 in Austria, with earlier presence retrospectively documented in Italy in 1996 [6]. Several African and European genotypes now circulate in Europe [7]. USUV was officially reported in eastern France (Haut-Rhin, Rhône and Bouches-du-Rhône departments) in 2015 by direct molecular identification in blackbirds [8] and mosquitoes [9], but its circulation in the country was suspected since 2009 [10]. While no human deaths have been attributed to this virus that is phylogenetically and ecologically close to WNV, USUV has caused several neuroinvasive disease cases in Europe recently. USUV has been found in blood donors, but transmission to recipients has not been documented [11].

The emergence of these two viruses always evolves towards a state of endemicity. Europe and the United States have witnessed WNV become endemic following initial outbreaks, with recurring cases linked to local transmission cycles that persist across seasons [17–19]. In France, both USUV (European and African genotypes) and WNV (lineages 1 and 2) are now circulating in endemic cycles. WNV and USUV had never been detected on the Atlantic coast of France before the end of summer 2022. Serological evidence of WNV circulation was reported in Nouvelle Aquitaine with the detection of an acute infection (presence of IgM and IgG specific antibodies) in 3 symptomatic horses in October 2022, coincident with a human case of USUV with no travel history outside the region. This date marked a turning point in the epidemiology of these viruses in France and foreshadowed an increase in cases the following year.

Both the high proportion of asymptomatic infections caused by WNV and USUV and their cryptic enzootic circulation make their detection in the environment challenging. There is currently no cost-effective and easy-to-use method for the early detection of the circulation of these viruses, which is essential for triggering the systematic viral RNA screening of blood and organ donors for these viruses.

Detecting arboviruses in field-collected mosquitoes is primarily used worldwide as an early warning system for viral circulation and to assess the risk of human transmission [20]. In most cases, arbovirus surveillance involves pooling mosquitoes by species, date, and location before extracting and detecting viral genomic RNA. For WNV surveillance, thousands to hundreds of thousands of mosquitoes are typically collected over a season, leading to pools of 50 to 200 mosquitoes per sample to minimize costs, with reported minimum infection rates ranging from less than 2% [21–24] to 10% [24,25] across studies. Here, at the interface between entomological and environmental surveillance, we have implemented a non-invasive molecular xenomonitoring (MX) approach that uses trapped mosquito excreta to monitor the emergence and circulation of WNV and USUV in real time.

The origins of the MX approach date back to the work of Hall-Mendelin and colleagues in 2010. The authors exploited mosquito sugar feeding to detect mosquito-borne pathogens in a community of trapped mosquitoes to improve the cost/effectiveness ratio of entomological surveillance (here defined as the detection of pathogens in mosquitoes) [26]. The discovery that the excreta of infected mosquitoes contain higher virus loads than their saliva [27], which thereby improve the sensitivity of molecular detection, led to a new arbovirus surveillance system that has proved its value in the field for several viruses [28,29] and parasites [30]. In MX, a 3D-printed housing that fits most standard mosquito traps, provides trapped mosquitoes with a moist shelter and freely accessible sugar water and facilitates the collection of their excreta on filter paper (S1 File). Trapped mosquitoes, here used as environmental samplers, are kept alive in the field over several days, which (i) allows the time between trap collections to be extended and (ii) increases the likelihood of virus shedding by trapped infected mosquitoes. Once infected by a virus, a single mosquito can shed between 3 to 5 log10 of viral RNA per day [27].

Excreta are shipped at ambient temperature by postal mail to a laboratory. There, these samples are used to detect viral RNA in a fast, simple, efficient, and cost-effective approach. Viral genetic identities can then be rapidly revealed by sequencing viral genomic RNA contained in excreta. Importantly, the method is compatible with downstream mosquito processing for nucleic acid extraction and sequencing, as well as virus isolation allowing to obtain viral genome sequences from individual mosquitoes, to identify vector species or to estimate mosquito infection prevalence (S1 Fig).

We demonstrated the value of MX in July-August 2023 in Nouvelle-Aquitaine, a region in southwest France with no prior history of WNV circulation until its initial detection in 2022.

## Materials and methods

### MX (Molecular Xenomonitoring) strategy

MX uses modified BG Sentinel traps (BGS, Biogents AG, Regensburg, Germany), inspired from Timmins et al. [31] and updated from L'Ambert et al. [28] (S2 Fig). In these modified BG Sentinel traps, BGS catching bags are replaced with a 3D printed MX adapter (S1 File) attached beneath the intake funnel through a conical net and inserted into the depressurized BGS catching pipe. The MX adapter provides a safe and moisturized shelter to trapped mosquitoes with easy access to a cotton ball soaked in 10% sugar water, held in a feeder at the top inner side of the cylinder. A filter paper (Whatman, grade 1, ref. 1001–917) is placed at the bottom of the adapter to collect excreta from trapped mosquitoes.

### Study area and samples collection

The study was carried out in a ~ 800 km$^2$ (40 x 20 km) area from either side of the Gironde estuary at the northern edge of the Bordeaux urban area, in the region of Nouvelle-Aquitaine (South-Western France) (Fig 1A). MX traps operated on a 24 hour/7-day basis using carbon dioxide ($CO_2$) as mosquito attractant. Pressurized $CO_2$ bottles operated at a rate of 250 mL/min, from 8 pm to 8 am using a BG-$CO_2$ timer (Biogents AG). Between 20th July and 3rd August 2023, four MX traps (A-D) were placed in a wetland area on the right bank of the Gironde estuary. Six MX traps (E-J) were placed in a wetland area between the Dordogne and the Garonne rivers, before their confluence. From August 11th, the MX surveillance was extended with 3 MX traps (K-M) located in an urban area, inside the city of Bordeaux. In sites A to J, captures were conducted over 3 to 4 consecutive days, with traps collection and reconditioning for a new mosquito trapping session performed twice a week. Trap reconditioning involves collecting the adapter with live mosquitoes and replacing it with a new one. K to M traps were emptied every 2 to 4 days. An additional mosquito sampling site was implemented at a late stage on the 10th of October 2023 in Châtelaillon, a town in the department of Charente-Maritime, which borders the Gironde department to the north. This sampling was carried out in the vicinity of a confirmed human case of West Nile virus.

Trapped mosquitoes were first stored at -20°C near the collection site, and then transferred to the laboratory where they were frozen at -80°C. The filter paper impregnated with mosquito excreta was removed and sent to the laboratory at room temperature by post. A new MX adapter was inserted in the trap for the following collection so that each trap was operating without discontinuity along the surveillance period.

### RNA extraction from filter papers impregnated with mosquito excreta

Filter papers impregnated with mosquito excreta were stored at 4°C upon arrival to the laboratory until the RNA extraction step. Samples were not frozen and were processed in less than

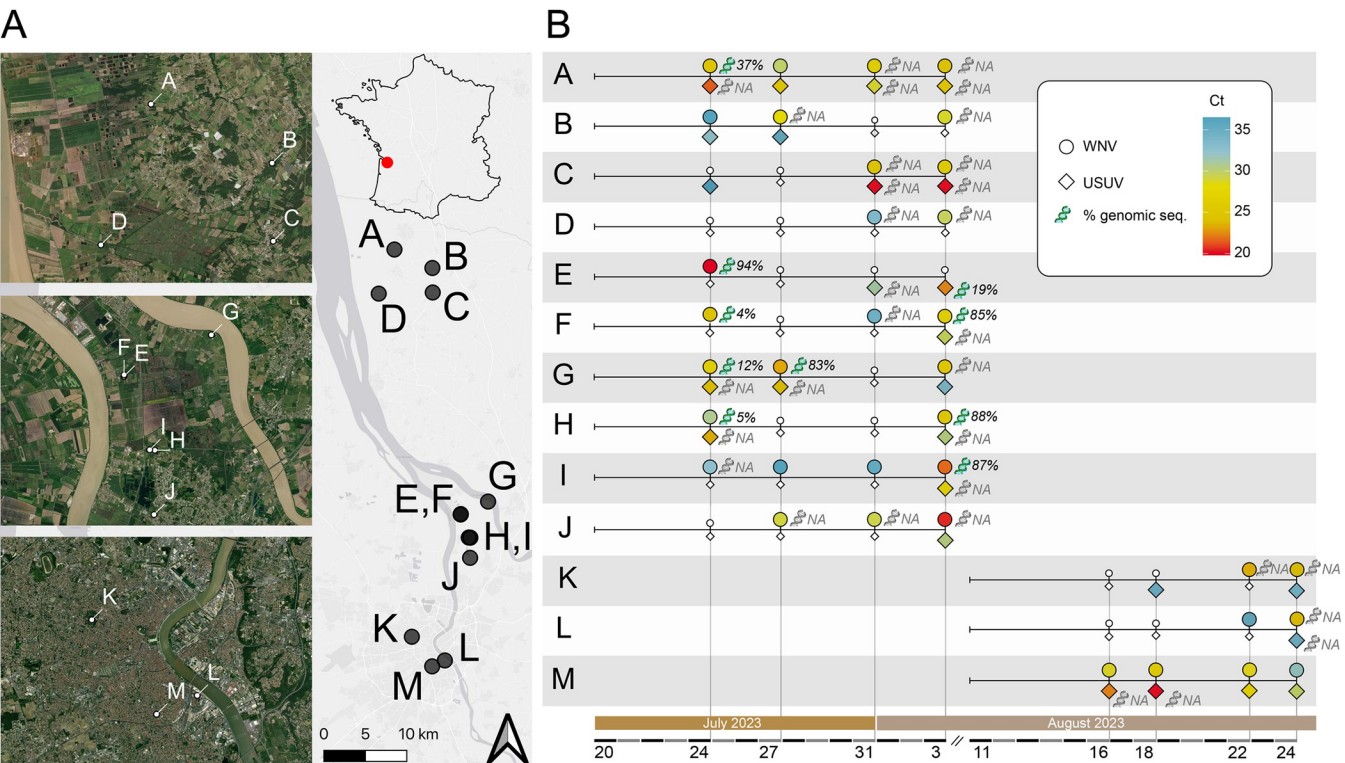

**Fig 1. Timeline of WNV and USUV RNA detection in trapped mosquito excreta from 13 sampling sites in the department of Gironde, region of Nouvelle-Aquitaine, in the South-West of France.** A) Study map with the geo-localization of the 13 sampling sites (A to M) situated on the East bank of the Gironde estuary, at the confluence of Dordogne and Garonne rivers, and within the Bordeaux agglomeration. The map was created using the free and open source QGIS geographic information system using satellite imagery from the ESRI. B) Detection of WNV (circles) and USUV (diamonds) in each sampling site (A to M) over time. Color shades correspond to cycle threshold (Ct) values, indicative of a virus load (red: low Ct and high virus load; blue: high Ct and low virus load). Little white symbols indicate no virus detection in mosquito excreta. The proportion of genomic sequence recovery from mosquito excreta at a site and collection time is represented by a DNA symbol with the genome coverage value at a sequencing depth of 50X. Grey DNA symbol indicates a failed attempt to generate sequence.

two weeks after reception. Briefly, each filter paper was coiled and placed at the bottom of a 14 mL plastic tubes before being soaked in 1.5 mL of Lysis buffer RAV1 (NucleoSpin 96 virus core kit, Macherey-Nagel, Düren, Germany) for 5 to 10 minutes. Ten microliters of MS2 phage were added to each tube as an internal extraction control [32]. Filter papers were then manually grinded with a 2 mL pipette until a homogeneous filter paper pulp was obtained. Several 2 mm diameter holes were then drilled in the transparent caps of the 14 mL tubes to create a colander and clipped tightly on each tube. Closed 14 mL tubes were then placed upside-down on a larger 50 mL tube (with the colander cap placed downward, at the bottom of the 50 mL tube) and centrifuged 5 min at 2,500 rpm. A 3D printed disposable spacer was used to create a space between the bottom of the 50 mL tube to avoid re-infiltration of the dry paper pulp by the flow through by capillarity after centrifugation. Flow throughs were collected and mixed with 1.5 mL 96–100% ethanol before being loaded on NucleoSpin Virus Columns in several steps. RNA extraction was then performed according to the manufacturer procedure. Eluates were stored at +4°C until used for molecular detection. This methodological step is summarized in S3 Fig.

### RNA extraction from individual mosquitoes

Using RT-qPCR results on mosquito excreta, we identified traps containing mosquitoes infected with WNV and/or USUV. RNA extraction and virus detection by RT-qPCR was

implemented individually for each mosquito from a selected subset of traps with positive RT-qPCR on excreta. The subset of traps were selected based on the following criteria: (*i*) the presence of a low number of trapped mosquitoes to reduce processing time, (*ii*) low *Ct* values obtained by RT-qPCR in excreta, indicative of a high virus load in trapped mosquitoes, (*iii*) a diversity of detection status in excreta with either the concomitant detection of both WNV and USUV, or one of these viruses alone, (*iv*) different location of traps (urban area, and rural area north of Bordeaux). These criteria were applied to increase the chance of virus isolation, to estimate the sensitivity and specificity of the method (working on excreta) as compared to standard entomological surveillance practices (working on mosquitoes), and to assess genomic diversity of strains circulating in different locations. Mosquitoes were individually homogenized using 3mm tungsten beads in 400 μL of Minimum Essential Medium (MEM) supplemented with 1% penicillin–streptomycin, 1% L-glutamine, 1% Kanamycin, and 3% Amphotericin B. Homogenization was realized in a 96-well plate in a TissueLyser grinder (QIAGEN, Hilden, Germany) for 2 × 30 seconds at 30 Hz. Viral RNA was extracted with QIAamp 96 Virus QIAcube HT Kit (QIAGEN, Hilden, Germany) on a QIAcube extraction platform using 100 μL of individual mosquito lysate. The remaining lysate volume was stored at 4°C for subsequent viral isolation attempts to be performed for RT-qPCR-positive mosquitoes.

## RT-qPCR for WNV and USUV

Detection of WNV and USUV genomic RNA was performed with a real-time reverse transcription polymerase chain reaction (RT-qPCR) assay. A dual-target in-house assay (USUV Duo) was used to detect USUV RNA. Two RT-qPCR assays were used to detect WNV RNA. A dual-target, single dye (FAM) RT-qPCR assay (Duo WNV), combining two assays from the literature [33,34], was used as a first-line due to its high sensitivity. Because this Duo assay cross-react with USUV, a second screening was performed with a WNV specific but less sensitive single target assay [34] in second intention to discriminate a single WNV infection from a co-infection with both viruses when needed. The SuperScript III Platinum One-Step qRT-PCR kit (ThermoFisher Scientific, Waltham, MA, USA) was used on a CFX96 thermal cycler, software version 3.1 (Bio-Rad Laboratories, Hercules, CA, USA). Cycling conditions were: 15 min at 50°C, 2 min at 95°C, 15 s at 95°C and 45 s at 60°C (45 cycles). A 5 μL volume of RNA was added to 20 μL of mix containing 12·5 μL of 2X Reaction Mix, 0·5 μL of Superscript III RT/Platinum Taq Mix and primers and probe at the concentrations described in S1 Table. A result was considered negative if the Ct value was > 40. The MS2 (internal control) RT-qPCR assay was tested on all samples.

## Viral isolation in cell culture

Remaining homogenates (kept at 4°C) were retrospectively selected from mosquitoes with a RT-qPCR result < 38 Ct, and inoculated on African green monkey kidney cells (Vero E6, ATCC C1008) and *Aedes albopictus* insect cells (C6/36, ATCC CRL-1660). Individual mosquito homogenates were filtered using 0·5 mL PVDF ST ultra-free-cl millipore (Merck, Darmstadt, Germany) and diluted (1/8) in 350 μL of MEM (for Vero E6 cells) or Leibovitz's L15 medium (for C6/36 cells) supplemented with 2.5% fetal bovine serum (FBS), 1% penicillin–streptomycin, 1% L-glutamine, 1% Kanamycin, and 3% Amphotericin B, before inoculation on confluent culture of Vero E6 and C6/36 on 6-well flat bottom cell culture plates. Individual mosquito homogenate inocula were incubated 1 hour at 37°C in a 5% CO2 atmosphere (Vero E6 cells) or 28°C without CO2 (C6/36 cells) to infect cells mono-layers prior to be removed and replaced by 4 mL of MEM (for Vero E6 cells) or L-15 (for C6/36 cells) supplemented with

7% heat-inactivated FBS, 1% penicillin–streptomycin, 1% L-glutamine, 1% Kanamycin, and 3% Amphotericin B. Cell cultures were examined daily for the presence of cytopathic effect (CPE). At day 5 post-inoculation, supernatants were aliquoted, RNA extracted and tested for the presence of WNV and USUV. Extraction and RT-qPCR were performed as described above.

Cultures were further screened with a sequence-independent method relying on the detection of dsRNA viral replicative intermediate. After a 5-day incubation, inoculated C6/36 or Vero E6 cells were transferred to four wells in 96-well plates then fixed using 150µL per well of 4% formaldehyde 0.5% TritonX-100 in PBS, incubated at 4˚C for 10 minutes. The fixative buffer was removed and the fixed cells were left to dry overnight. We then performed an ELISA using the monoclonal antibody to viral RNA intermediates in cells (MAVRIC) 2G4 as primary antibody, as per O'Brien et al. 2021 [35,36].

## Virus sequencing

For sequencing WNV and USUV genomes from virus isolates, cell culture supernatants were treated with benzonase for 1h at 37˚C, extracted with no RNA carrier, and a random RT-PCR amplification was performed using the TransPlex Complete Whole Transcriptom Amplification Kit WTA2 (Merck Millipore) following the manufacturers' instructions. Following amplification (virus-specific or random), an equimolar pool of all amplicons was prepared for each sample, and then purified and quantified before being sonicated into 250 pb long fragments. Fragmented DNA was used for library building followed by PCR quantification. Finally, an emulsion PCR of the library pools was performed, followed by loading on 530 chips and sequencing using the S5 Ion torrent technology, following the manufacturer's instructions.

We sequenced WNV and USUV genomes from excreta and whole mosquito samples using an amplicon-based approaches. We used virus-specific sets of primers to generate eight overlapping amplicons spanning the entire virus genome (S1 Table). Virus-specific sets of primers (S1 Table) were used to generate eight overlapping amplicons spanning the entire virus genome with the Superscript IV one step RT-PCR System (ThermoFisher Scientific). Briefly, PrimalSeq protocol generates overlapping amplicons from 2 multiplexed PCR reactions to generate sufficient templates for subsequent high-throughput sequencing. We used primer schemes developed for WNV lineage 2 [37] and USUV [38]. PCR mixes (final volume 25µL) contained 3µL of nucleic acid extract, 1,25µL of each primer (10µM), 12.5 µL of 2X Platinum SuperFi RT-PCR Master Mix, 6.5µL of RNAse free water and 0.5µL of SuperScript IV RT Mix. Amplifications were performed using the following conditions: 10 min at 55˚C, 2 min at 98˚C, followed by 40 cycles with the three following steps: 10 sec at 98˚C, 10 sec at 55˚C and 1.45 min at 68˚C, and a final step at 68˚C for 5 min.

When amplification failed for one or more amplicons, we later attempted whole genome amplification using a tiled-amplicon approach initially developed by Quick J. [39], and Grubaugh N.D.[40] and colleagues, with adaptations. We attempted whole genome amplification using a tiled-amplicon approach by generating 20 (WNV) or 35 (USUV) overlapping amplicons from 2 multiplexed PCR reactions with the Superscript IV one step RT-PCR System (ThermoFisher Scientific). PCR mixes (final volume 25µL) contained 3µL of nucleic acid extract, 1,25µL of each primer (10µM), 12.5 µL of 2X Platinum SuperFi RT-PCR Master Mix, 6.5µL of RNAse free water and 0.5µL of SuperScript IV RT Mix. Amplifications were performed using the following conditions: 10 min at 55˚C, 2 min at 98˚C, followed by 40 cycles with the three following steps: 10 sec at 98˚C, 10 sec at 55˚C and 1.45 min at 68˚C, and a final step at 68˚C for 5 min.

After demultiplexing, read data were analyzed with an in-house Snakemake pipeline. Reads were first trimmed using cutadapt (v4.4) to remove amplification primers (virus-specific

amplification approach) and with trimmomatic (v0.39) to remove short and low quality reads. Read alignment was achieved using BWA MEM (v0.7.17) using, as a reference, the best match identified by blasting (magicblast, v1.7.7) sequencing reads using a database of flavivirus sequences including reference sequences representative of the genetic diversity of WNV (8 sequences) and USUV (13 sequences). Consensus sequences were called using the ivar (v1.3.1) consensus command, and a minimum coverage depth of 50x (virus-specific amplification approach) or 30x (random amplification approach). Regions with insufficient coverage were masked with N characters.

## Phylogenetic analyses

All publicly available sequences for WNV and USUV were downloaded from the NCBI Nucleotide database, Genbank (database accessed on November 17nd, 2023). We filtered the data by: (i) excluding sequences from laboratory strains (adapted, passaged multiple times, obtained from experiments), (ii) keeping only sequences covering more than 85% of the open reading frame (ORF). The remaining sequences were trimmed to their ORF, aligned using MAFFT (version 7.511) and inspected manually using the program AliView (version 1.0). We inferred the phylogenetic relationships between public WNV and USUV genomes and the sequences generated in this study based on a Maximum-likelihood (ML) approach with IQ-Tree (version 1.6.12), using the best-fit model identified by ModelFinder and assessed branch support using an ultrafast bootstrap approximation (UFBoot2) (1000 replicates). Based on these first phylogenies, we selected smaller datasets including, for WNV, all sequences from the Central/South-West European subgroup of lineage 2 (267 sequences) and for USUV, all sequences from the Africa 3 genotype (128 sequences). Using these datasets we reconstructed time-scaled phylogenies with BEAST (v1.10.5), under the Shapiro-Rambaut-Drummond-2006 (SRD06) substitution model an uncorrelated lognormal (UCLN) clock model clock, and a bayesian skygrid coalescent models. We ran five MCMC chains of 50 million states with the BEAGLE computational library. We used Tracer (v1.7) for inspecting the convergence and mixing, discarding the first 10% of steps as burn-in, and ensuring that estimated sampling size (ESS) values associated with estimated parameters were all >200. All xml files for these analyses are available at https://github.com/rklitting/WNV_USUV_NouvelleAquitaine_2023. Phylogenetic trees were visualized using the ggtree R package.

## Mosquito species identification through mitochondrial DNA

Molecular identification at the species level was performed on all mosquitoes with a PCR-positive result for either WNV or USUV, and mosquitoes that failed to be identified at the species level morphologically. Morphological identification was confirmed at the species level for a subset of mosquitoes by sequencing a 710 bp region of the cytochrome oxidase subunit I (COI) using primers from Simon et al. [41] (S1 Table). Five μl of mosquito excreta RNA/DNA eluates were used in a 20 μl PCR mix containing 5 μl of Hot START 5X Firepol ready-to-load DNA polymerase mix (Dutscher, Brumath, France), 2 μl of forward and reverse primers at 10 μM, and 11 μl of water. The thermal programme was: 10 min of polymerase activation at 96˚C followed by 35 cycles of (i) 30 s denaturing at 96˚C, (ii) 30 s annealing at 50˚C and (iii) 1 min extension at 72˚C, followed by a final incubation step at 72˚C for 7 min to complete synthesis of all PCR products. Amplicons were subsequently sequenced using the Sanger method and the reverse primer at Microsynth AG, Lyon, France. Each sequence was visually inspected and compared with nucleotide sequences database deposited in GenBank using the BLAST algorithm (S3 File).

### Digested blood meal identification using amplicon-based metabarcoding on mosquito excreta

A ~440 bp mitochondrial DNA section corresponding to a subfragment of COI was amplified with primers developed by Reeves et al. [42] (S1 Table) using RNA/DNA eluates extracted from trapped mosquito excreta. These primers were degenerated to selectively amplify vertebrate mitochondrial DNA while avoiding co-amplification of mosquito mitochondrial DNA. Illumina Nextera universal tails sequences were added to the 5′ end of each of these primers to facilitate library preparation by a two-step PCR approach. Six nucleotides barcodes were also inserted in the reverse primer sequences to reduce the costs by multiplexing [43]. The parameters for PCR mix and cycling were the same as for mosquito species identification described above. A 15 cycle PCR was then performed using Nextera Index Kit–PCR primers, that adds the P5 and P7 termini that bind to the dual 8 bp index tags and the flow cell. Resulting amplicons were purified with magnetic beads (SPRIselect, Beckman Coulter). Libraries were sequenced on a MiSeq run (Illumina) by Microsynth AG, Zurich, Switzerland, using MiSeq version 3 chemistry with 300 bp paired-end sequencing.

The DDemux program [43] was used for demultiplexing fastq files. Demultiplexed.fastq sequences were imported to QIIME 2 Amplicon Distribution version 2023.9 for bioinformatic analyses. The qiime2-dada2 pipeline [44] was used for turning paired-end fastq files into merged reads, filtering out Illumina adapters, denoising and removal of chimeras and filtering out replicates. Taxonomic assignment was carried out for the amplicon sequence variants (ASVs) using the qiime2-feature-classifier classify-consensus-vsearch plugin using a database of 1,176,764 sequences gathering Fungi, Protist, and Animal COI records, recovered from the Barcode of Life Database Systems 7 March 2021 available in L'Ambert et al.[28]. Phylogenetic tree was made with the iqtree-ultrafast-bootstrap function implemented in QIIME 2 based on sequences (ASV). The script and input QIIME 2 artifacts are provided in S2 File.

### Statistical analyses and data visualization

All statistical analyses were performed in the statistical environment R. Figures were made using the package ggplot2 (Wickham, 2016). The Map was created using the Free and Open Source QGIS Geographic Information System using satellite imagery from the Environmental Systems Research Institute (ESRI). All other base map layers and choropleths were created in R using the ggplot2 package.

## Results

### WNV and USUV were detected in trapped mosquito excreta at high rate

A total of 52 excreta samples, obtained from 13 different sites in the region of Nouvelle-Aquitaine between the end of July and August 2023, were collected and processed (Fig 1). WNV or USUV RNA was detected in 39 (75%) excreta, alone or in combination. WNV RNA was detected in 35/52 (67%) filters. USUV RNA detected in 26/52 (50%) filters. WNV and USUV RNA were both detected in the same filter in 22/52 cases (42%). At least one of the two viruses was detected at least once in all 13 sites over the study period, Site D (rural area in the north of Bordeaux) is the only site where WNV alone was detected. There was no site with an exclusive detection of USUV. WNV was detected at a late stage in an additional trap in Charente Maritime.

### WNV and USUV detection in trapped mosquitoes

Of the 39 filter-positive traps, we selected 7 traps (18%) to test captured mosquitoes, totalizing 364 mosquitoes. These mosquito collections originated from 6 sites at 5 different dates (site E:

**Table 1. Concordance of WNV and USUV detection in trapped mosquitoes and their excreta.** Only mosquitoes from a subset or traps with PCR-positive results on excreta (7 collections, C1-C7) have been tested by RT-qPCR.

| Collection | Sampling date | Sites | Location | Virus detection in excreta | N mosquitoes | *Culex* | *Aedes* | Others | N (%) WNV | N (%) USUV |
|---|---|---|---|---|---|---|---|---|---|---|
| C1 | 25/07/2023 | E | Estuary | WNV | 28 | 28 | 0 | 0 | 5 (18%) | 0 (0%) |
| C2 | 03/08/2023 | E | Estuary | USUV | 60 | 58 | 2 | 0 | 0 (0%) | 5 (8%) |
| C3 | 03/08/2023 | G | Estuary | WNV+USUV | 35 | 32 | 3 | 0 | 6 (17%) | 0 (0%) |
| C4 | 03/08/2023 | I | Estuary | WNV+USUV | 27 | 19 | 5 | 3 | 0 (0%) | 0 (0%) |
| C5 | 03/08/2023 | F | Estuary | WNV+USUV | 26 | 17 | 9 | 0 | 2 (8%) | 0 (0%) |
| C6 | 18/08/2023 | M | City | USUV | 94 | 83 | 4 | 7 | 0 (0%) | 4 (4%) |
| C7 | 24/08/2023 | K | City | WNV+USUV | 94 | 55 | 32 | 7 | 5 (5%) | 2 (2%) |
| | | | | **Total** | **364** | **292** | **55** | **17** | **18 (5%)** | **11 (3%)** |

25/07/2023 and 03/08/2023; sites G, I and F: 03/08/2023; site M: 18/08/2023; site K 24/08/2023). Each of the 364 mosquito was tested individually to estimate positivity rates. The positivity rate in mosquitoes from different traps ranged from 5% (N = 5/94) to 18% (N = 5/28) for WNV and from 2% (N = 2/94) to 8% (N = 5/60) for USUV (Table 1). Considering all mosquito samples, the overall positivity rate by PCR was 5% and 3% for WNV and USUV, respectively. These results may overestimate positivity rates, as we only tested mosquitoes from traps for which a PCR-positive result was obtained for WNV or USUV in the corresponding excreta. Each of the 364 mosquito was morphologically identified at the genus level: 291 (80%) and 51 (14%) were classified as *Culex* (*Cx.*) and *Aedes* (*Ae.*)/*Ochlerotatus*, respectively. The morphologically unidentified mosquitoes were subjected to molecular identification (COI sequencing). COI was also sequenced in all mosquitoes that were PCR positive for either WNV or USUV (N = 17) to identify the species.

*Cx. pipiens* accounted for 77% of infected mosquitoes (N = 23/30): 14 were positive for WNV and 9 for USUV. Notably, no other *Culex* species were successfully identified at the molecular level among the PCR-positive mosquitoes. Among non-*Culex* mosquitoes that were found positive for either viruses, molecular identification revealed *Ochlerotatus caspius* (N = 2, positive for WNV), *Aedes vexans* (N = 2, positive for WNV), *Culiseta longiareolata* (N = 1, positive for USUV), and *Aedes albopictus* (N = 1, positive for WNV) (S4 File). The total diversity of mosquito species captured and molecularly identified in this study was limited to 5 species: *Cx. pipiens*, *Oc. caspius*, *Ae. vexans*, *Ae. albopictus* and *Cs longiareolata*. *Cx. pipiens* was the most frequently captured species. It was the only species–together with *Ae. albopictus*–that was trapped in every locations, both urban and rural. *Cs longiareolata* was the only species caught only in the urban environment.

*Cx. pipiens* showed both the highest PCR positivity rate and viral loads, independently of the virus. WNV was detected in *Cx. pipiens* with a mean Ct value of 29·8 (range 15·2–37·6 Ct), and in *Aedes/Ochlerotatus* with a mean Ct value of 33·9 (range min 33·1–34·3 Ct). USUV was detected in *Cx. pipiens* with Ct value as low as 18·8 (range 18·8–39·9 Ct, mean: 34·5 Ct), in *Cs longiareolata* with a Ct of 17·2 and in an undetermined *Aedes* species with a Ct of 39·3.

In collections C1, C2, C6 and C7, perfect agreement was observed between the presence of viral RNA in excreta and in the corresponding mosquitoes (Table 1). However, no USUV RNA was detected in mosquitoes from collections C3, C4 and C5, although viral RNA from both viruses had previously been detected in excreta.

A total of 34 *Culex* spp., 79 *Aedes* spp. and 7 other mosquitoes non morphologically identified at the genus level were collected in the trap from Charente Maritime in October. WNV was detected with a Ct of 35·4 in the pool of *Culex* mosquitoes (S3 File).

### High success of viral isolations from single mosquitoes with the help of data obtained from trapped mosquito excreta

All individual mosquito homogenates with a Ct < 38 were selected for attempting virus isolation. Four WNV and three USUV strains were isolated from individual mosquitoes. The success rate of virus isolation from a single mosquito was 4/18 (22%) and 3/7 (43%) for WNV and USUV, respectively; it was linked to virus load with 6/7 (86%) isolates coming from mosquitoes with a Ct<21 (S2 Table). All strains were isolated on both Vero E6 and C6/36 cells, except for one USUV strain (Isolate numb. 7), which was recovered on Vero E6 cells. All PCR positive virus isolates were also positive both Vero E6 and C6/36 cells with a sequence-independent ELISA method on relying on the detection of dsRNA viral replicative intermediate [35]. This method also detected viruses in additional samples negatives for WNV or USUV by RT-qPCR in C6/36 insect cells only, suggesting the presence of insect-specific viruses.

### Virus sequencing on excreta and mosquitoes confirmed the presence of WNV or USUV and revealed their genetic identities

To confirm the presence of WNV and USUV, we sequenced virus genomes from positive excreta and mosquitoes using an amplicon approach. WNV genomes were found in 8 of 26 mosquito excreta samples (35%) and USUV genomes in 1 of 17 samples (6%). For individual mosquitoes, we sequenced 4 WNV and 2 USUV genomes, and from isolates, we obtained 5 WNV and 3 USUV genomes. All WNV genomes were Lineage 2, and all USUV genomes were Africa 3 genotype.

Phylogenetic analysis of WNV genomes from Nouvelle-Aquitaine revealed a distinct monophyletic clade within lineage 2, closely related to strains from Austria, Slovakia, the Czech Republic, and Germany (2014–2020) (Central/South-West European subgroup [45], Figs 2A, 2D and S4). This clade is not directly related to the L1 and L2 WNV previously found in French animals.

For USUV, genomes from Nouvelle-Aquitaine grouped with a 2018 sequence from Haute-Vienne within the Africa 3 genotype (Fig 2B and 2E). Similar results were obtained from virus genomes in mosquito excreta (S5 Fig). This clade is rooted in sequences from Germany, Belgium, and the Netherlands (2016–2018), indicating a single main lineage circulating in the region.

### Sequencing digested blood from trapped mosquitoes reveals a community of vertebrates exposed to mosquitoes

DNA amplification was successful in 9 (17%) samples, all from the rural sites B (27/07/2023), D (27/07/2023), E (25 and 27/07/2023), F (27/07/2023), G (25/07/2023), H (25 and 27/07/2023) and I (25/07/2023). Sequencing generated a total of 244,424 demultiplexed read sequences across all samples with a Q20 of 92% and a median of 9,950 read sequences per sample. The total number of reads was reduced to 75 amplicons sequence variants (ASVs) with a mean length of 381 nucleotides, a mean occurrence of 1·7 ASV per sample (min: 1; first quartile: 1; third quartile: 1; max: 8) and a mean frequency of 633X (min: 2X; first quartile: 6X; third quartile: 43·5X; max: 23,879X) per sample. Following taxonomic assignment, a total of 11 ASVs (15%) were assigned to the *Arthropoda* phylum with *Chironomus riparius*, *Ochlerotatus detritus*, *Phlebotomus perniciosus*, and *Hybrizon buccatus* identified at the species level. A majority of ASVs (N = 56%) were unassigned using our identity threshold, and 18 (24%) were assigned to the *Chordata* phylum. Among them 13 (17%) were identified as human DNA. As we cannot exclude that the presence of human DNA is due to contamination during sample

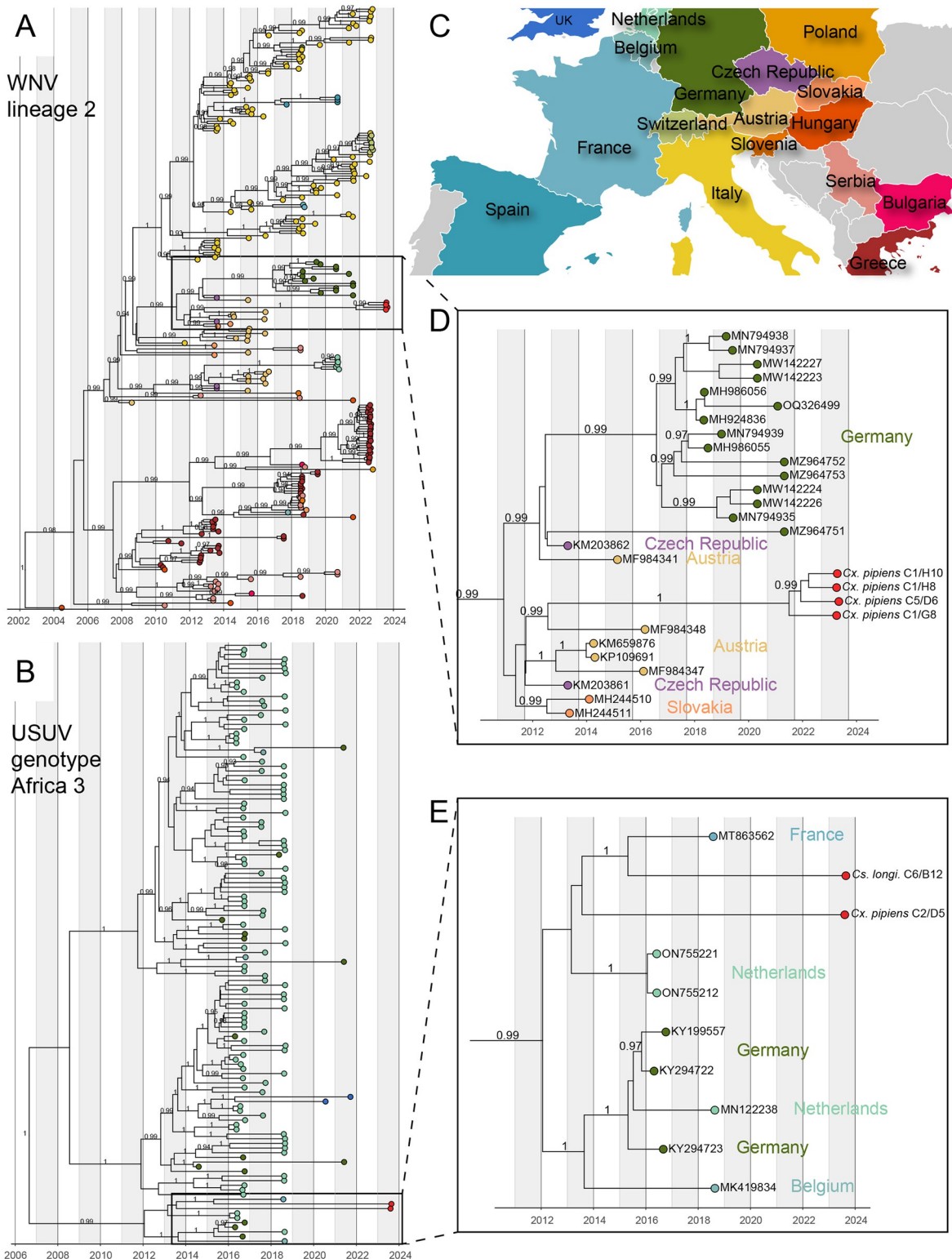

**Fig 2. Phylogenetic relationships within WNV lineage 2 (Central European/Hungarian clade) and USUV Africa 3 genotype with a focus on sequences from Nouvelle-Aquitaine.** Maximum clade credibility trees for WNV and USUV time-scaled phylogenies were reconstructed with BEAST (v1.10.5). Clade posterior supports superior to 90% are shown. A) Phylogenetic relationships within WNV from lineage 2 Central European/Hungarian clade are represented. All sequences are colored according to their geographic origin. B) Phylogenetic relationships within USUV Africa 3 genotype are represented. All sequences are colored according to their geographic origin.

C) Map color highlighting the color-country correspondence used for tip coloring. D) and E) Zoom on the phylogenetic clades corresponding to WNV (D) and USUV (E) sequences from Nouvelle-Aquitaine. Map base layers are from the R maps package available at CRAN: Package maps (r-project.org).

processing, these ASVs were not considered here. Twelve ASVs were assigned to 10 vertebrate taxons: *Bos taurus* (cow), *Sus scrofa* (pig), *Canis lupus* (dog or wolf), *Felis catus* or *Felis silvestris* (cat), *Equus caballus*, *Equus ferus* or *Equus przewalskii* (horses) and *Rattus rattus* (rat) from the *Mammalia* class, *Podarcis muralis* (common wall lizard) from the *Reptilia* class, and *Streptopelia decaocto* (Eurasian collared dove), *Gallus gallus* (hen) and *Milvus migrans* or *Milvus milvus* (kites) from the *Aves* class (Fig 3). Hens, Cats, and dogs were the vertebrate species that were the most identified in both occurrence (number of study sites) and quantity (ASV counts) in these samples.

## Discussion

### A need for a new surveillance method to early detect cryptic enzootic arbovirus circulation

The accidental transmission from an organ or blood donor to recipients is one of the main risks associated with the silent circulation of enzootic neurotropic arboviruses such as WNV and, to a lesser extent, USUV. In France, systematic viral screening of all blood and organ donors is currently triggered by the diagnosis and reporting of incident WNV human cases. There is no systematic surveillance system for USUV in humans. Animal surveillance for WNV and USUV is mainly based on the reporting and diagnosis of sick horses (WNV) and dead birds (WNV, USUV). Due to the large proportion of asymptomatic infections, neither method are fully effective in detecting the circulation of WNV/USUV in their enzootic cycles

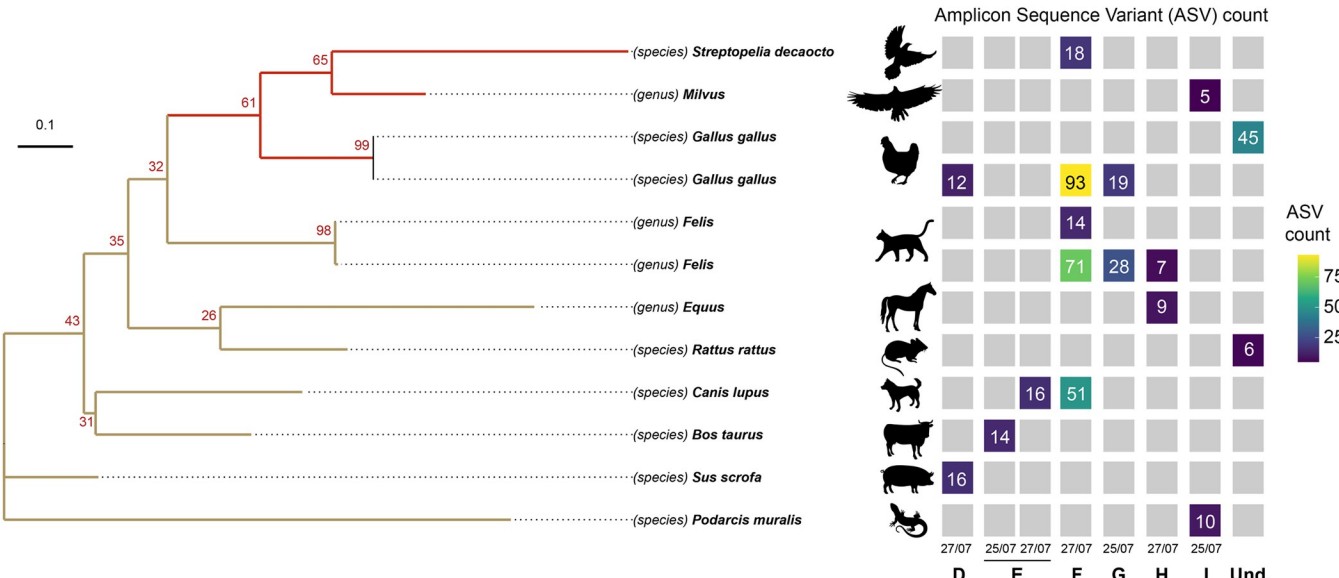

**Fig 3. Vertebrate diversity identified based on digested blood from trapped mosquitoes.** Taxons were determined at the species or genus level by comparing ASV to a sequence database using the vsearch algorithm. Genus level was chosen when an ASV matches several species inside a genus using our parameters. *Milvus*: *Milvus migrans* and *Milvus milvus*, *Felis*: *Felis catus* and *Felis silvestris*, *Equus*: *Equus caballus*, *Equus ferus* or *Equus przewalskii*. The molecular phylogenetic tree was created with the iqtree-ultrafast-bootstrap function implemented in QIIME 2 directly from ASV sequences, with 100 bootstrap replicates. This phylogenetic tree is therefore not necessarily representative of the genetic distances between these taxons. Heatmap represents the total number of ASV attributed to a taxon according to sampling sites and sampling times.

before they cause disease in humans and animals. Entomological surveillance offers non-invasive early warning capabilities, but this approach is expensive and labor intensive, requiring the processing of large numbers of mosquitoes by trained operators with strong entomological skills [46].

Xenomonitoring broadly refers to the detection of vector-borne human pathogens in blood-feeding arthropod vectors. As the name suggests, it enables ongoing surveillance of pathogens in the environment without the need for human testing. One of the earliest formal applications of xenomonitoring was for lymphatic filariasis (LF), where it helped health programs assess the success of mass drug administration (MDA) campaigns aimed at eliminating the disease [47]. Initially reliant on observing parasites in dissected insects under a microscope, the concept evolved into Molecular Xenomonitoring (MX) with the advent of molecular detection methods, which enhanced sensitivity and improved large-scale screening by detecting pathogens at the species level in pools of captured arthropods [48]. MX is now used for a range of mosquito-borne parasites and viruses, as well as for pathogens transmitted by other vectors [47]. Here, we extend the noninvasive concept of MX (i.e., external molecular surveillance) by detecting viruses directly in the excrement of mosquitoes collected in the field.

## MX reveals the hidden circulation of enzootic arboviruses in Nouvelle-Aquitaine

Here, MX succeeded in detecting WNV and USUV enzootic transmissions in Nouvelle-Aquitaine with a high rate of detection in trapped mosquito excreta (75% of samples), one year after their suspected emergence in this area, as evidenced by serological detection in equids. In less than 2 weeks following collection, WNV and USUV genomes were detected and sequenced directly from the RNA extracted from the mosquito excreta. MX detected WNV concurrently with the confirmation of the first WNV human case in the region (end of July 2023) and a few days before the first equine case in Gironde department (4th of August). A total of 22 and 4 confirmed human cases were subsequently reported during the transmission season for WNV and USUV, respectively. MX revealed the hidden circulation of USUV concomitantly with the first confirmed human case and before the first avian case (end of July and mi-August respectively). In Italy, entomological surveillance was reported to be able to outpace the appearance of the first human infections by days to weeks [49,50]. Its early implementation in the season via MX can thus be effective in detecting enzootic arboviral circulation in endemic or emerging areas while viruses are still invisibly amplifying in animal reservoirs [51]. The presence of WNV and USUV in Gironde, as revealed by human and animal surveillance, has made it possible to extend the detection of viral genomes in donations of human products to the surrounding Charente Maritime department from 2023 August 10th. MX, which was carried out late at the vicinity of a WNV human case in this department, showed that WNV was still present in the environment two months later.

MX has demonstrated many advantages over traditional entomological surveillance: (i) screening mosquito excreta is time and cost efficient, as one sample is tested for viral RNA, regardless of the number and species diversity in the trap; (ii) excreta can be preserved and transported from the field to the laboratory at room temperature by regular postal mail (S4 File), bringing real-time detection and genetic identification within reach; (iii) the method increases the longevity of trapped mosquitoes, thereby allowing extension of the time between trap collections and increasing the likelihood of virus shedding by infected mosquitoes. MX is easy to implement in the field and requires neither a strong entomological background nor specific technical skills. Virus RNA preservation is central to the MX method. In a spiking and recovery experiment using *Aedes flavivirus*, an insect specific flavivirus with similar

morphologic and genomic characteristics than WNV, we demonstrated that dried viral RNA remained detectable by RT-qPCR across varied support materials and temperatures up to one month, even at 37˚C (S4 File).

## *Culex* mosquitoes were major vectors in the transmission of WNV and USUV in Nouvelle-Aquitaine

*Cx. pipiens* was the most abundant species collected in both urban and rural environments and the species found with both the highest infection rate and viral loads. This species unquestionably played a leading role in the transmission of both WNV and USUV viruses in Nouvelle-Aquitaine, as it has already been reported in other transmission areas [52,53]. Here, WNV was also detected in other mosquito species. While the detection of viral RNA in mosquitoes does not necessarily confirm their infection or their ability to transmit the virus—they may simply have fed on viraemic hosts without being infected—the high USUV load recovered from a *Culiseta longiareolata* suggests a role for this ornithophilic species in the enzootic amplification cycle of these viruses. In addition, the detection of WNV in *Ae. albopictus* echoes recent findings that this urban and anthropophilic mosquito species is able to transmit WNV and USUV under experimental conditions [54]. Unlike *Culex* mosquitoes, this invasive species has been reported to have strict mammalian orientated feeding preferences [55] and an avian component is a prerequisite for amplification of these viruses prior to infection of mosquitoes. The potential role of this species in bridging the animal reservoir to human hosts, alongside *Culex* mosquitoes, may warrant further attention if occasional blood-feeding on birds can occur.

Here, both WNV and USUV were mostly isolated from *Culex* mosquitoes exhibiting high virus loads. Infected arthropods are prime targets for virus isolation because, unlike vertebrates, they do not develop sterilizing immunity and can amplify viruses throughout their lives. Isolating viruses as they evolve and emerge worldwide can feed research activities to better understand or forecast the mechanisms that underlie their spread, pathogenicity, or adaptability in new environments.

## Easy and early access to arbovirus genomic sequences shed light on the origin and spread of viruses

Here, we obtained virus sequences directly from trapped mosquito excreta, rapidly classified the circulating WNV and USUV strains as belonging to lineage 2 and Africa 3, respectively, and confirmed these results using virus sequences obtained from single individuals. Virus sequencing from excreta samples may provide useful elements to quickly assess the potential origin of viral circulation. The resulting virus consensus genomes constitute, however, a mixing of virus populations from all virus-excreting mosquitoes–when more than one are present in the trap. In that case, analysing such genomes using phylogenetic methods may not be accurate. For that reason, we performed phylogenetic inference using virus genomes from individual mosquitoes (rather than excreta) to try to trace the origin of the virus strains identified in this study (Fig 2).

While the first detection of WNV circulation in France dates back to the 1960s [56], limited sequence data are available to assess the spatio-temporal dynamics of circulation of the virus in the country. Here, we show that WNV sequences originating from the Nouvelle-Aquitaine region are distinct from previous L2 sequences identified in the South of France (Alpes maritimes) in 2018 from dead raptors specimens [16] (S4 Fig) and group with sequences from Austria. Based on our phylogenetic inference, the most recent common ancestor between Nouvelle-Aquitaine and Austrian sequences is approximately 10 years old (Fig 2), which makes it difficult to identify the actual timing and geographical source of the introduction of

WNV into South-West France. The latter might be in Austria but may alternatively be located in another unsampled country closer to France including Italy, which has already been identified as a likely source of WNV introduction in the past [16].

The detection of USUV in France is more recent than for WNV and dates back to 2015, when the virus was detected in the North-East (Haut-Rhin, Rhône) and South of France (Camargue), with distinct virus lineages circulating in each location, those from Rhin being apparently related to German sequences (Europe 3 genotype), those from Rhone appearing closer to sequences from Spain (Africa 2 genotype), and those from Camargue being closer to sequences from Germany and Spain (Africa 2 genotype), and the Netherlands (Africa 3 genotype). In 2018, the USUV Africa 3 genotype was identified again in Haute-Vienne, this last virus sequence is the closest phylogenetic relative of the USUV sequences identified in this study, with whom it shares a most recent common ancestor around 10 years ago (Fig 2). The long branches linking those events of virus circulation in 2018 and 2023 suggest that an important unsampled diversity of USUV genotype Africa 3 circulates in Nouvelle Aquitaine.

Altogether, our results highlight our limited knowledge of the circulating genetic diversity of WNV and USUV in France and in Europe. They call for increased genomic surveillance of arboviruses to (i) improve our understanding of the spatio-temporal circulation dynamics of these viruses at a large scale, (ii) better predict the sequential expansion of the viruses beyond the borders of Nouvelle Aquitaine and (iii) better inform public health strategies, in particular, vector management interventions.

### Molecular information contained in digested blood meals can help to reveal ecological factors involved in the emergence of these viruses

The ecological factors underlying the emergence of WNV and USUV in a Nouvelle-Aquitaine region remain unresolved. A link with the migration of birds, which are reservoirs for these viruses, can reasonably be suggested. The large fires south of Bordeaux in 2022 may also have destroyed a natural buffer zone and displaced bird populations. Mosquito excreta contains digested blood that the mosquitoes have ingested from the surrounding fauna before being caught. Sequencing regions of the selected vertebrate portion of this DNA mixture can provide insight into their local trophic preference. MX has the asset to capture vertebrate blood as it is progressively digested by trapped mosquitoes, without the need to process the mosquitoes during the digestion stage. While it cannot directly identify an animal reservoir, it does link a diversity of trapped mosquitoes to a diversity of surrounding animal hosts and, when applied at scale and combined with viral genetic information, can help to reveal the ecological forces at play in the emergence and transmission dynamics of these viruses.

### Limitations

In this study, as in previous one [28], viral RNA was detected in the excreta of trapped mosquitoes, even when none was found in the mosquitoes themselves. Although we cannot completely rule out the possibility of mosquitoes escaping during sampling or trap handling, the more plausible explanation is the presence of predatory arthropods, such as ants and spiders, using the traps as food storage. Ants, in particular, probably first attracted by the sugar source inside the adapters, have been observed severing mosquitoes' legs or wings and carrying them back to their nests, leaving detached mosquito parts in the collections. This recurring issue, particularly associated with long-term field captures over several days, remains difficult to resolve. Integrated approaches that leverage site-specific solutions can help mitigate this issue, such as suspending traps from a glue-covered wire or placing them over a basin of water.

Excreta-based contamination of viral RNA on the body surface of mosquitoes trapped alongside infected individuals has been observed in other organisms [57], suggesting this possibility cannot be ruled out here. Such contamination could lead to an overestimation of arbovirus prevalence using the MX method and may complicate the identification of mosquito vector species involved in local transmission. Determining the vector competence of mosquitoes in the field is challenging, regardless of the trapping method used. While dissecting individual mosquitoes can provide insights, it is impractical for large-scale screening. Field studies have typically reported WNV infection rates in *Culex* mosquitoes ranging from 0.05% to 8%, depending on factors such as location, environmental conditions, and proximity to local bird reservoirs. [58,59]. Our estimate falls within this typical range, indicating consistency with broader findings on WNV prevalence in mosquito populations. High viral loads detected in individual mosquito homogenates can help identify the mosquito vector species responsible for local transmission. While more field-based observations are essential for understanding the role of local mosquito species in WNV or USUV transmission, vector competence assays in controlled experimental settings can further clarify this issue [60,61].

## Conclusion

A major drawback of entomological surveillance is that it requires time and specific knowledge. Combined with the low infection rates that typically occur in low or non-endemic areas, the method can have an unfavorable cost/effectiveness ratio that can hinder its promotion in nationwide arbovirus surveillance programs with steady funding and operational commitment. Here, we implemented a non-invasive, innovative, efficient, and cost-effective MX approach, at the crossroads between entomological and environmental surveillance, that succeeded in revealing the hidden circulation and the genetic identity of WNV and USUV in Nouvelle-Aquitaine. By taking advantage of excreta-based PCR testing, the MX strategy significantly accelerates the identification of potential infection hotspots, streamlining the surveillance process and facilitating more rapid and targeted public health mitigation and control measures.

## Supporting information

**S1 Table. Molecular amplification systems used in this study.** Oligonucleotides sequences (primers and probes) are presented with their corresponding species, gene targets and amplicon sizes. Illumina Nextera universal tails sequences that have been added to primers during the PCR amplification step of the metabarcoding method are represented in green and barcodes in blue. An adenine (A) nucleotide was added between the barcode and the primer (not mandatory).
(XLSX)

**S2 Table. Ct values obtained from excreta, individual mosquito samples and Vero E6 and C6/36 supernatant samples obtained for isolated WNV and USUV strains.** Collection/Position field correspond to the first and second column of supplementary file 3 related to virus screening in individual mosquitoes.
(DOCX)

**S3 Table. WNV and USUV sequences produced from mosquito excreta and individual mosquitoes.** For each sequence, the exact source (excreta, mosquito, isolate), amplification approach, coverage at 30X (for sequences obtained from isolates) or 50X (for sequences obtained from mosquito and mosquito excreta samples), and Genbank accession number are specified. Sequencing reads for all virus genomes are available on NCBI (Bioproject ID:

PRJNA1085973). Virus genomes with no Genbank accession number (NA*) are available at https://github.com/rklitting/WNV_USUV_NouvelleAquitaine_2023.
(DOCX)

**S4 Table. Primer sequences used to generate a non-overlapping amplicon ladder of different sizes to assess viral RNA degradation.**
(DOCX)

**S1 Fig. Workflow of the MX approach.** Mosquitoes are captured and kept alive on the field during several days in a 3D printed shelter with a free access to sugar water. Mosquitoes are then killed and kept frozen *in situ* while the filter papers containing their excreta are sent to a laboratory at room temperature by post. Virus detection is performed at first step directly on mosquito excreta by RT-qPCR. If positive, an attempt was made to sequence the genomic RNA of the virus using amplicon-based approaches directly on the excreta. Mosquitoes from collections found positive for either viruses were then transported to the laboratory on dry ice before to be analyzed individually. Estimation of infection rates in mosquitoes, virus isolation and sequencing were performed on trapped mosquitoes on a second step.
(TIF)

**S2 Fig. Representations of the 3D printed MX adapter designed to increase trapped mosquitoes' longevity and to collect their excreta for an arbovirus surveillance purpose.** (A) Different views of the adapter. All components are visible in the cross-sectional and exploded views. (B) Picture of the adapter ready to be attached to the intake funnel of the BGS. The MX adapter was created on Fusion 360 (AutoDesk) and 3D printed in PLA. MX adapter 3D files (.stl format) are provided in S1 File under the Creative Commons (CC) license BY.
(TIF)

**S3 Fig. Schematic representation of the procedure to extract RNA/DNA from filter papers impregnated with mosquito excreta.**
(TIF)

**S4 Fig. Phylogenetic relationships with WNV species with a focus on sequences from Nouvelle-Aquitaine obtained from excreta, single mosquito extract and cell culture isolates.** The Maximum-likelihood phylogeny was inferred using IQ-Tree under model finder. Branch support values were calculated using UFBoot (100 replicates). Statistical supports values superior to 80% are shown for the main clades. All sequences are colored according to their geographic origin. A zoom on the clade with WNV sequences from this work is shown on the right hand side of the panel (Excreta: sequences derived from mosquito excreta, VEROE6 and C636: sequences derived from VEROE6 and C636 cell cultures, respectively, Mosquito: sequences derived from single mosquitoes). Map base layers are from the R maps package available at CRAN: Package maps (r-project.org).
(TIF)

**S5 Fig. Phylogenetic relationships with USUV species with a focus on sequences from Nouvelle-Aquitaine obtained from excreta, single mosquito extract and cell culture isolates.** The Maximum-likelihood phylogeny was inferred using IQ-Tree under model finder. Branch support values were calculated using UFBoot (100 replicates). Statistical supports values superior to 80% are shown for the main clades. All sequences are coloured according to their geographic origin. A zoom on the clade with WNV sequences from this work is shown on the right hand side of the panel (Excreta: sequences derived from mosquito excreta, VEROE6 and C636: sequences derived from VEROE6 and C636 cell cultures, respectively, Mosquito: sequences derived from single mosquitoes). Map base layers are from the R maps package

available at CRAN: Package maps (r-project.org).
(TIF)

**S6 Fig. Schematic representation of the procedure used to assess the Molecular Detection Sensitivity in Preserved Dried Flavivirus RNA.** In this design, the volume of PBS was dependent of the support material.
(TIF)

**S7 Fig. Virus detection as cycle threshold (Ct) based on time post-inoculation, temperature, and support material.** Mean (±SD) Ct values across replicates are presented in an heatmap.
(TIF)

**S8 Fig. Agarose gel electrophoresis of PCR-amplified AEFV genomic regions of varying lengths to evaluate viral genomic RNA degradation under different conditions.** Viral RNA was exposed to various temperatures, support materials, and time points post-inoculation. Amplicons were visualized on a 1% agarose gel, providing insights into RNA integrity and stability across experimental conditions.
(TIF)

**S1 File. MX adapter 3D files in.stl format.** MX adapter is under the Creative Commons (CC) license BY-NC-SA (Licensees may copy, distribute, display, and make derivatives only for non-commercial purposes and by giving credits to the authors). Two versions are provided. Version 1 was used in this work. Version 2 is updated to decrease the cost and printing time.
(RAR)

**S2 File. Archive comprising (i) Qiime2 code that was used in the amplicon-based metabarcoding analysis pipeline (Script_qiime2_dada2.sh), (ii) metadata file associated to the data (COI-metadata.txt), (iii) input reads (COI-paired-end.qza).** All analysis and diversity metrics implemented in Qiime2 can be accessed by running Qiime2 diversity commands on data provided. All.qsv files generated by the script can be easily loaded on the Qiime2 visualizer at https://view.qiime2.org.
(RAR)

**S3 File. Data relative to virus detection in collections of trapped individual mosquitoes and their molecular identification at the species level.**
(XLSX)

**S4 File. Evaluation of molecular detection sensitivity in preserved dried *Flavivirus* RNA: effects of time, temperature, and virus Dose.** This supplementary file present results of a spiking and recovery experiment that aim to assess how different transportation and storage conditions impact the molecular detection sensitivity of *Flavivirus* RNA. The materials and methods, as well as the results and discussion, are provided in this file.
(DOCX)

## Acknowledgments

We thank Thomas Canivez, Laurent Bosio, Manon Geulen, and Manon Peden from the CNR des arbovirus, Sophie Lescure, Christophe Courtin, Steeve Vernede, Hadrien Martin-Herrou from Bordeaux Métropole, Guéric Gabriel and Franck Bastit from the Communauté de Commune de Blaye and Benoît Leuret, Frédéric Jacquet and Léonard Bour from the Direction départementale de la protection des populations (DDPP) de la Gironde, Sebastien Chouin, Laurent Malnoe, Yann Renaudeau, Christian Smeraldi, and Stephane Macaud from la

direction de l'Environnement et de la mobilité du département de la Charente Maritime for their help and technical assistance. The PCR tests were provided by the European virus archive-Marseille (EVAM, https://evam.european-virus-archive.com/) under the label Technological Platforms of Aix-Marseille.

## Author Contributions

**Conceptualization:** Clément Bigeard, Grégory L'Ambert, Gaëlle Gonzalez, Alexandre Duvignaud, Denis Malvy, Xavier de Lamballerie, Albin Fontaine.

**Formal analysis:** Raphaelle Klitting.

**Investigation:** Clément Bigeard, Laura Pezzi, Raphaelle Klitting, Nazli Ayhan, Grégory L'Ambert, Nicolas Gomez, Géraldine Piorkowski, Rayane Amaral, Agathe M. G. Colmant, Cynthia Giraud, Katia Ramiara, Camille Migné, Thierry Touzet, Denis Malvy, Xavier de Lamballerie, Albin Fontaine.

**Methodology:** Clément Bigeard, Laura Pezzi, Raphaelle Klitting, Nazli Ayhan, Nicolas Gomez, Géraldine Piorkowski, Rayane Amaral, Guillaume André Durand, Agathe M. G. Colmant, Camille Migné, Gilda Grard, Thierry Touzet, Stéphan Zientara, Rémi Charrel, Gaëlle Gonzalez, Denis Malvy, Xavier de Lamballerie, Albin Fontaine.

**Resources:** Grégory L'Ambert, Guillaume André Durand, Katia Ramiara, Gilda Grard, Thierry Touzet, Stéphan Zientara, Rémi Charrel.

**Supervision:** Clément Bigeard, Thierry Touzet, Stéphan Zientara, Gaëlle Gonzalez, Alexandre Duvignaud, Denis Malvy, Xavier de Lamballerie, Albin Fontaine.

**Validation:** Rémi Charrel, Alexandre Duvignaud, Denis Malvy, Xavier de Lamballerie, Albin Fontaine.

**Visualization:** Raphaelle Klitting, Albin Fontaine.

**Writing – original draft:** Clément Bigeard, Laura Pezzi, Raphaelle Klitting, Nazli Ayhan, Gaëlle Gonzalez, Alexandre Duvignaud, Denis Malvy, Xavier de Lamballerie, Albin Fontaine.

**Writing – review & editing:** Agathe M. G. Colmant, Rémi Charrel, Gaëlle Gonzalez, Alexandre Duvignaud, Denis Malvy, Xavier de Lamballerie, Albin Fontaine.

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
