## [Decision Letter · Decision Letter 0]

10 Oct 2024

Dear Mr. Fontaine,

Thank you very much for submitting your manuscript "Molecular Xenomonitoring (MX) allows real-time surveillance of West Nile and Usutu virus in mosquito populations" for consideration at PLOS Neglected Tropical Diseases. As with all papers reviewed by the journal, your manuscript was reviewed by members of the editorial board and by several independent reviewers. The reviewers appreciated the attention to an important topic. Based on the reviews, we are likely to accept this manuscript for publication, providing that you modify the manuscript according to the review recommendations. 

The reviewers provided suggestions on improving minor issues with the manuscript (writing, typos, grammar, etc). The main critique was shared by multiple reviewers and the authors should adapt their manuscript accordingly in response to this critique. Namely, the authors should relate their "new" method explicitly to "older" established methods. This could be accomplished by experimentation (suggested by one reviewer, but not necessary, in the opinion of the editor). As suggested by some reviewers, this ideally would be addressed by including more discussion of the results of the current manuscript relative to (1) prior manuscripts on the topic of xenomonitoring and (2) more classical methods of vector surveillance. In other words, make a stronger case for xenomonitoring by explicitly comparing it to classical methods, including direct comparison of the observed results to expectations based on classical surveillance and previous findings of xenomonitoring. This was in the manuscript already to a limited extent, but the reviewers request a more obvious/stronger comparison.

Sincerely,

Jeremy V. Camp, Ph.D.

Academic Editor

Elvina Viennet

Section Editor

The reviewers provided suggestions on improving minor issues with the manuscript (writing, typos, grammar, etc). The main critique was shared by multiple reviewers and the authors should adapt their manuscript accordingly in response to this critique. Namely, the authors should relate their "new" method explicitly to "older" established methods. This could be accomplished by experimentation (suggested by one reviewer, but not necessary, in the opinion of the editor). As suggested by some reviewers, this ideally would be addressed by including more discussion of the results of the current manuscript relative to (1) prior manuscripts on the topic of xenomonitoring and (2) more classical methods of vector surveillance. In other words, make a stronger case for xenomonitoring by explicitly comparing it to classical methods, including direct comparison of the observed results to expectations based on classical surveillance and previous findings of xenomonitoring. This was in the manuscript already to a limited extent, but the reviewers request a more obvious/stronger comparison.

Reviewer's Responses to Questions

**Key Review Criteria Required for Acceptance?**

**Methods**

-Are the objectives of the study clearly articulated with a clear testable hypothesis stated?

-Is the study design appropriate to address the stated objectives?

-Is the population clearly described and appropriate for the hypothesis being tested?

-Is the sample size sufficient to ensure adequate power to address the hypothesis being tested?

-Were correct statistical analysis used to support conclusions?

-Are there concerns about ethical or regulatory requirements being met?

Reviewer #1: This is a very good manuscript. The methodological strategy is innovative with an interesting perspective.

Reviewer #2: (No Response)

Reviewer #3: Methods could be compared to more standard ones. See main review for details.

Reviewer #4: Objectives clear and operational relevance obvious

Study design is appropriate

Methods and design clearly articulated and illustrated

Sample size small but sufficient for this "proof of principle"

Analyses are apporpriate.

No ethics required.

**Results**

-Does the analysis presented match the analysis plan?

-Are the results clearly and completely presented?

-Are the figures (Tables, Images) of sufficient quality for clarity?

Reviewer #1: Results are clear with enough quality

Reviewer #2: (No Response)

Reviewer #3: (No Response)

Reviewer #4: Results are comprehensive and well illustrated.

**Conclusions**

-Are the conclusions supported by the data presented?

-Are the limitations of analysis clearly described?

-Do the authors discuss how these data can be helpful to advance our understanding of the topic under study?

-Is public health relevance addressed?

Reviewer #1: Conclusion are correct but they missed the part that the molecular biology techniques requiere infrastructure. The next step is to identify infected vectors with straigthforward staregies that can be applied in the local areas without sending any material for processing.

Reviewer #2: (No Response)

Reviewer #3: Yes

Reviewer #4: The data supports the conclusions and the application and operational / public health relevance is clear.

The authors could have tried to think of some limitations, and given the claims that the method is highly resource and cost effective, it woudl have been nice to see some numbers, however roughly generated, incom prison with alternative approaches.

**Editorial and Data Presentation Modifications?**

Reviewer #1: None

Reviewer #2: (No Response)

Reviewer #3: (No Response)

Reviewer #4: (No Response)

**Summary and General Comments**

Reviewer #1: Good manuscript with interesting perspectives.

Reviewer #2: The molecular xenomoniroing (MX) approach proposed in this study seems a very innovative and promising strategy to simplify mosquito borne pathogen surveillance.

The origins of this approach date back to two previous works (Hall-Mendelin et al. 2010 and Fontaine et al. 2016) in which the authors exploited mosquito saliva and excreta collected on filter papers to detect mosquito-borne pathogens. In MX approach, a 3D-printed housing that fits most standard mosquito traps, was adopted to facilitates the collection of their excreta on filter paper improving the cost/effectiveness ratio of entomological surveillance.

The manuscript is very well written, with carefully described objectives and methods. Results are clearly and properly presented.

My suggestions are as follows:

Line 77-84: can be deleted. Line 70-76 and 77-84 repeat the same thing

Line 383-384: it would be better to write the genus in full for all species (Ochlerotatus caspius, Ades vexans , Culiseta longiareolata and Aedes albopictus)

Line 386: Ochlerotatus caspius and Culiseta longiareolata can be shortened in Oc. caspius and Cs. longiareolata 

Line 514-518: A total of 19/364 mosquito specimens positive to WNV are reported in this study (Cx. pipiens N=14, Ochlerotatus caspius N=2, Aedes vexans N=2, Aedes albopictus N=1). More than one fourth of the positive sample is represented by species occasionally found in nature positive for this virus, with unclear or unproven vector role (e.g. Ochlerotatus caspius, Aedes vexans, Aedes albopictus). According to the authors, it is plausible that such positivity in non-vector species could be determined by a blood meal on viraemic hosts without being infected. This finding could be explained by high viral circulation in their hosts and is also suggested by the high number of positive excreta samples (39/52). In my opinion, positivity in mosquitoes should be better investigated by extending the analysis to a larger number of samples to strengthen the finding of high viral circulation (at least in their vectors) and exclude contamination in non-vector species.

Reviewer #3: The manuscript “Molecular Xenomonitoring (MX) allows real-time surveillance of West Nile and Usutu virus in mosquito populations” presented by Clément Bigeard and colleagues provides an insightful analysis of the status of WNV and USUV activity in Atlantic seaboard of France and a novel means of monitoring viral activity. In this manuscript the authors convey:

• Sufficient sensitivity and specificity of XM in detecting WNV and USUV in mosquito excreta from various collection sites

• Optimal isolation and purification of virus from individual mosquitoes from trapped mosquito excreta

• Identification and confirmation of vertebrate species impacted by mosquito feeding and potential implications

I was especially impressed with the demonstrated ease and efficiency of XM for WNV and USUV and am curious as to its application on other arboviruses and arthropod species. While I believe that this manuscript provides valuable insight into a novel means of surveillance for relevant pathogens of concern, there are areas of concern in its current form. The manuscript would greatly benefit from comparison of the proposed methods to standard practices to better demonstrate its advantages. While this may be challenging, my view is that it would increase the impact of the paper if it can be done.

Overall this is an interesting paper that will be of interest to the readership of PNTD. It is generally well done and the analyses appear to have been conducted correctly. 

Specific Comments on the manuscript follow: 

1. Please define WNV and USUV in the abstract prior to the author summary (line 29).

2. “MX can early detect” is confusing (line 44).

3. Please reference Box 1 to end of first paragraph in introduction (line 69). 

4. “The emergence of these two viruses always evolves towards a state of endemicity” comes across as an overestimation. Please provide references that supports this (line 123). 

5. The introduction would benefit from the inclusion of a more detailed explanation of currently early detection methods. 

6. The manuscript would benefit from the inclusion of a more detailed explanation of the geography of the study area. Specifically, providing background about Nouvelle-Aquitaine’s prior history of mosquito or arbovirus activity would help convey to the reader how the study was being executed (line 154-155). 

7. Is it known what the stability in storing the mosquito excreta-impregnated filter paper at room temperature? Is virus degradation observed and too what degree?

8. “The presence of a low lumber”? Please confirm is lumber is correct (line 203).

9. Please include a separate section discussing statistical tests used (line 342-344). 

10. Please provide a justification or reasoning behind collecting from these sites at different timeframes in Figure 1 (line 356-366).

11. There should be discussion on why no USUV RNA was detected even though it was found in excreta as a potential limitation of XM (line 399-400).

12. This is a great demonstration of MX use and capabilities (line 502-508)

13. Have the authors completed a side-by-side comparison of detecting virus in diluted mosquito excreta to determine the level of detection?

Reviewer #4: This is an extremely well-written, well-structured, and carefully analysed paper. MX is highly topical, as is the difficulty in defining transmission pathways for arboviral zoonoses with many vectors and reservoirs. The authors used a nice combination of PCR, viral isolation and sequencing to deliver robust conclusions about the utility of this excreta-based, MX method. Mosquito IDs and vertebrate hosts were identified through the incorporation of conventional barcoding. 

The authors should be congratulated on delivery of an excellent manuscript and for making an important contribution to the field.

PLOS authors have the option to publish the peer review history of their article (what does this mean?). If published, this will include your full peer review and any attached files.

Reviewer #1: Yes: Humberto Lanz-Mendoza

Reviewer #2: No

Reviewer #3: No

Reviewer #4: No

Figure Files:

Data Requirements:

Reproducibility:

References

---

## [Decision Letter · Decision Letter 1]

2 Dec 2024

Dear Mr. Fontaine,

We are pleased to inform you that your manuscript 'Molecular Xenomonitoring (MX) allows real-time surveillance of West Nile and Usutu virus in mosquito populations' has been provisionally accepted for publication in PLOS Neglected Tropical Diseases.

Best regards,

Jeremy V. Camp, Ph.D.

Academic Editor

Elvina Viennet

Section Editor

Shaden Kamhawi

co-Editor-in-Chief

Paul Brindley

co-Editor-in-Chief

Reviewer's Responses to Questions

**Summary and General Comments**

Reviewer #2: The authors responded satisfactorily to the reviewers' comments, fully discussing the results and clarifying some critical issues. The manuscript is overall improved and can be accepted.

Reviewer #3: The authors have responded adequately to my prior critique. I have no further comments on this manuscript.

PLOS authors have the option to publish the peer review history of their article (what does this mean?). If published, this will include your full peer review and any attached files.

Reviewer #2: No

Reviewer #3: No

---

## [Editor Report · Acceptance letter]

17 Dec 2024

Dear Mr. Fontaine,

We are delighted to inform you that your manuscript, "Molecular Xenomonitoring (MX) allows real-time surveillance of West Nile and Usutu virus in mosquito populations," has been formally accepted for publication in PLOS Neglected Tropical Diseases.

Best regards,

Shaden Kamhawi

co-Editor-in-Chief

Paul Brindley

co-Editor-in-Chief
